# Efficient and Robust Spiking Neural Circuit for Navigation Inspired by Echolocating Bats

**Pulkit Tandon, Yash H. Malviya**
Indian Institute of Technology, Bombay
`pulkit1495,yashmalviya94@gmail.com`

**Bipin Rajendran**
New Jersey Institute of Technology
`bipin@njit.edu`

## Abstract

We demonstrate a spiking neural circuit for azimuth angle detection inspired by the echolocation circuits of the Horseshoe bat *Rhinolophus ferrumequinum* and utilize it to devise a model for navigation and target tracking, capturing several key aspects of information transmission in biology. Our network, using only a simple local-information based sensor implementing the cardioid angular gain function, operates at biological spike rate of approximately $10\,\mathrm{Hz}$. The network tracks large angular targets ($60°$) within 1 sec with a $10\%$ RMS error. We study the navigational ability of our model for foraging and target localization tasks in a forest of obstacles and show that it requires less than 200X spike-triggered decisions, while suffering less than $1\%$ loss in performance compared to a proportional-integral-derivative controller, in the presence of $50\%$ additive noise. Superior performance can be obtained at a higher average spike rate of $100\,\mathrm{Hz}$ and $1000\,\mathrm{Hz}$, but even the accelerated networks require 20X and 10X lesser decisions respectively, demonstrating the superior computational efficiency of bio-inspired information processing systems.

## 1 Introduction

One of the most remarkable engineering marvels of nature is the ability of many species such as bats, toothed whales and dolphins to navigate and identify preys and predators by echolocation, i.e., emit sounds with complex characteristics, and use neural circuits to discern the location, velocity and features of obstacles or targets based on the echo of the signal. Echolocation problem can be sub-divided into estimating range, height and azimuth angle of objects in the environment. These coordinates are resolved by the bat using separate mechanisms and networks [1, 2]. While the bat's height detection capability is obtained through the unique structure of its ear that creates patterns of interference in the spectrum of incoming echoes [3], the coordinates of range and azimuth are estimated using specialized neural networks [1, 2].

Artificial neural networks are of great engineering interest, as they are suitable for a wide variety of autonomous data analytics applications [4]. In spite of their impressive successes in solving complex cognitive tasks [5], the commonly used neuronal and synaptic models today do not capture the most crucial aspects of the animal brain where neuronal signals are encoded and transmitted as spikes or action potentials and the synaptic strength which encodes memory and other computational capabilities is adjusted autonomously based on the time of spikes [6, 7]. Spiking neural networks (SNNs) are believed to be computationally more efficient than their second-generation counterparts[8].

Bat's echolocation behavior has two distinct attributes – prey catching and random foraging. It is believed that an 'azimuth echolocation network' in the bat's brain plays a major role in helping it to forage randomly as it enables obstacle detection and avoidance, while a 'range detection network' helps in modulating the sonar vocalizations of the bat which enable better detection, tracking and catching of prey [1, 2]. In this paper, we focus on the relatively simple azimuth detection network of the greater horseshoe bat to develop a SNN for object tracking and navigation.

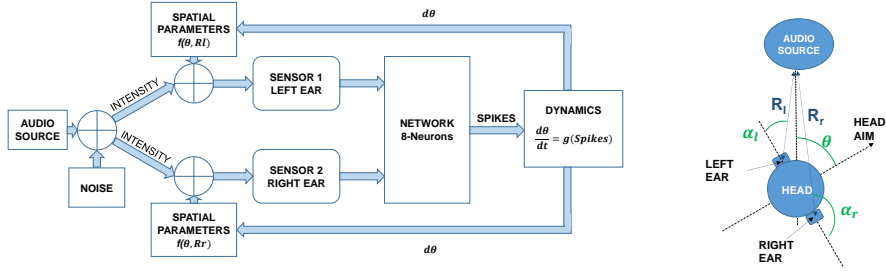

Figure 1: Schematic diagram of the navigation system based on a spiking neural network (SNN) for azimuth detection, inspired by bat echolocation. The two input sensors (mimicking the ears), encode incoming sound signals as spike arrival rates which is used by the SNN to generate output spikes that controls the head aim. Spikes from the top channel induces an anti-clockwise turn, and the bottom channel induces a clockwise turn. Thus, the head-aim is directed towards maximum intensity. We use small-head approximation (i.e., $R_l = R_r$, $\alpha_l = \pi/2 - \theta$ and $\alpha_r = \pi/2 + \theta$).

## 2   System Design

We now discuss the broad overview of the design of our azimuth detection network and the navigation system, and the organization of the paper. The functioning of our echolocation based navigation model can be divided into five major parts. Figure 1 illustrates these parts along with a model of the tracking head and the object to be detected. Firstly, we assume that all objects in the environment emit sound isotropically in all our simulations. This mimics the echo signal, and is assumed to be of the same magnitude for simplicity. Let the intensity of an arbitrary source be denoted as $I_s$. We assume that the intensity decays in accordance with an inverse square dependence on the distance from the source. Hence, the intensity at the ears (sensors) at a distance $R_l$ and $R_r$ will be given as

$$I_l = \frac{I_s}{R_l^2} \qquad\qquad I_r = \frac{I_s}{R_r^2} \qquad (1)$$

The emitted sound travels through the environment where it is corrupted with noise and falls on the receivers (bat's ears). In our model, the two receivers are positioned symmetric to the head aim, $180°$ apart. Like most mammals, we rely on sound signal received at the two receivers to determine azimuth information [1]. By distinguishing the sound signals received at the two receivers, the network formulates a direction for the potential obstacle or target that is emitting the signal.

In our model, we use a cardioid angular gain function as input sensor, described in detail in Section 3. We filter incoming sound signals using the cardioid, which are then translated to spike domain and forwarded to the azimuth detection network. The SNN we design (Section 4) is inspired by several studies that have identified the different neurons that are part of the bat's azimuth detection network and how they are inter-connected [1], [2]. We have critically analyzed the internal functioning of this biological network and identified components that enable the network to function effectively.

The spiking neural network that processes the input sound signals generates an output spike train which determines the direction in which the head-aim of our artificial bat should turn to avoid obstacles or track targets. The details of this dynamics are discussed in Section 5. We evaluate the performance of our system in the presence of ambient noise by adding slowly varying noise signals to the input (section 6). The simulation results are discussed in Section 7, and the performance of the model evaluated in Section 8, before summarizing our conclusions.

## 3   Input and Receiver Modeling

The bat has two ears to record the incoming signal and like most mammals relies on them for identifying the differences at these two sensors to detect azimuth information [1]. These differences could either be in the time of arrival or intensity of signal detected at the two ears. Since the bat's head is small (and the ears are only about $1 - 2\,\text{cm}$ apart) the interaural time difference (ITD), defined as the time difference of echo arrival at the two ears, is very small [9]. Hence, the bat relies on measurement of the interaural level difference (ILD), also known as interaural intensity difference

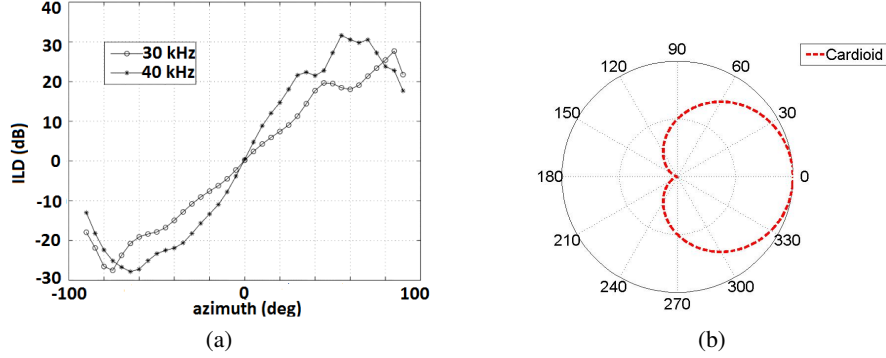

Figure 2: a) The Interaural Level Difference, defined as relative intensity of input signals received at the two sensors in the bat is strongly correlated with the azimuth deviation between the sound source and the head aim. Adapted from [9]. b) In our model, sensitivity of the sensor (in dB) as a function of the angle with source obeys cardioid dependence (readily available in commercial sensors).

(IID) for azimuth angle detection. As shown in Figure 2a, the ILD signal detected by the ears is a strong function of the azimuth angle; our network is engineered to mimic this characteristic feature.

In most animals, the intensity of the signal detected by the ear depends on the angle between the ear and the source; this arises due to the directionality of the ear. To model this feature of the receiver, we use a simple cardioid based receiver gain function as shown in Figure 2b, which is the most common gain characteristic of audio sensors available in the market. Hence, if $\alpha_{r/l}$ is the angle between the source and the normal to the right/left ear, the detected intensity is given as

$$I_{d,r/l} = I_{r/l} \times 10^{-1+cos(\alpha_{r/l})} \tag{2}$$

We model the output of the receiver as a spike stream, whose inter-arrival rate, $\lambda$ encodes this filtered intensity information

$$\lambda_{r/l} = kI_{d,r/l} \tag{3}$$

where $k$ is a proportionality constant chosen to ensure a desired average spiking rate in the network. We chose two different encoding schemes for our network. In the uniform signal encoding scheme, the inter-arrival time of the spikes at the output of the receiver is a constant and equal to $1/\lambda$. In the Poisson signal encoding scheme, we assume that the spikes are generated according to a Poisson process with an inter-arrival rate equal to $\lambda$. Poisson source represents a more realistic version of an echo signal observed in biological settings. In order to update the sound intensity space seen by the bat as it moves, we sample the received sound intensity ($I_{d,r/l}$) for a duration of 300 ms at every 450 ms. The fallow 150 ms between the sampling periods allows the bat to process received signals, and reduces interference between consecutive samples.

## 4 Spiking Neural Network Model

Figure 3a shows our azimuth detection SNN inspired by the bat. It consists of 16 sub-networks whose spike outputs are summed up to generate the network output. In each sub-network, Antroventral Cochlear Nucleus (AVCN) neurons receive the input signal translated to spike domain from the front-end receiver as modeled above and deliver it to Lateral Superior Olive (LSO) neurons. Except for the synaptic weights from AVCN layer to LSO layer, the 16 sub-networks are identical. The left LSO neuron further projects excitatory synapses to the right Dorsal Nucleus of Lateral Lemniscus (DNLL) neuron and to the right Inferior Colliculus (IC) neuron and inhibitory synapses to the left DNLL and IC neurons. Additionally, inhibitory synapses connect the DNLL neurons to both IC neurons which also inhibit each other. The AVCN and IC neurons trigger navigational decisions.

Minor variations in the spike patterns at the input of multi-layered spiking neural networks could result in vastly divergent spiking behaviors at the output due to the rich variations in synaptic dynamics. To avoid this, we use 16 clone networks which are identical except for the weights of synapse from AVCN layer to LSO layer (which are incremented linearly for each clone). These clones operate in

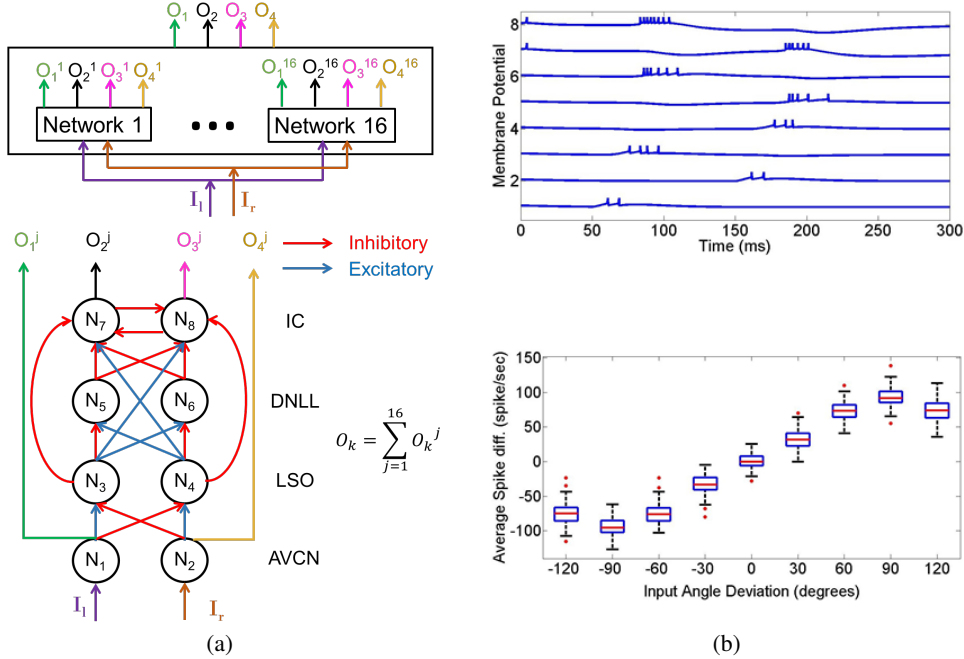

$$O_k = \sum_{j=1}^{16} O_k{}^j$$

Figure 3: (a) Our azimuth detection SNN consists of 16 sub-networks whose spike outputs are summed up to generate the network output. Except for the synaptic weights from AVCN layer to LSO layer, sub-networks are identical (see Supplementary Materials). Higher spike rate at the left input results in higher output spike rate of neurons $N_1$ and $N_8$. (b) Top panel shows normalized response of the SNN with impulses presented to input neuron $N_1$ at $t = 50$ ms and $N_2$ at $t = 150$ ms. Bottom panel shows that the output spike rate difference of our SNN mimics that in Figure 2a.

parallel and the output spike stream of the left and right IC neurons are merged for the 16 clones, generating the net output spike train of the network.

We use the adaptive exponential integrate and fire model for all our neurons as they can exhibit different kinds of spiking behavior seen in biological neurons [10]. All the neurons implemented in our model are regular spiking (RS), except the IC layer neurons which are chattering neurons (CH). CH neurons aggregate the input variations over a period and then produce brisk spikes for a fixed duration, thereby improving accuracy. The weights for the various excitatory and inhibitory synapses have been derived by parameter space exploration. The selected values enable the network to operate for the range of spike frequencies considered and allows spike responses to propagate through the depth of the network (All simulation parameters are listed in Supplementary Materials).

An exemplary behavior of the network corresponding to short impulses received by $AVCN_{left}$ at $t = 50$ ms and by $AVCN_{right}$ at $t = 150$ ms is shown in Figure 3b. If $\lambda_{right}$ of a particular sound input is higher than $\lambda_{left}$, the right AVCN neuron will have a higher spiking rate than its left counterpart. This in turn induces a higher spiking rate in the right LSO neuron, while at the same time suppressing the spikes in left LSO neuron. Thereafter, the LSO neurons excite spikes in the opposite DNLL and IC neurons, while suppressing any spikes on the DNLL and IC neurons on its side. Consequently, an input signal with higher $\lambda_{right}$ will produce a higher spike rate at the left IC neuron.

It has been proposed that the latter layers enable extraction of useful information by correlating the past input signals with the current input signals [11]. The LSO neuron that sends excitatory signals to an IC neuron also inhibits the DNLL neuron which suppresses IC neuron. Inhibition of DNLL neuron lasts for a few seconds even after the input signal stops. Consequently, for a short period, the IC neuron receives reduced inhibition. Lack of inhibition changes the network's response to future input signals. Hence, depending on the recent history of signals received, the output spike difference may vary for the same instantaneous input, thus enabling the network to exhibit proportional-integral-derivative controller like behavior. Figure 4 highlights this feature.

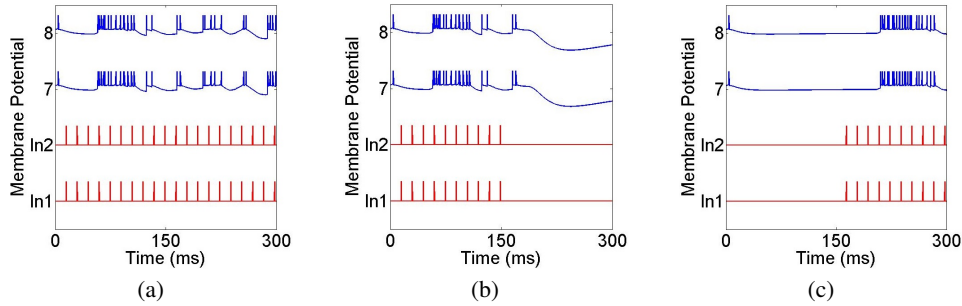

Figure 4: Spike response of the network (blue) depends not only on the variations in the present input (red) but also on its past history, akin to a proportional-integral-derivative controller. Input spike trains are fed to neurons $N_1$ and $N_2$. Choosing the input spikes in (a) as reference, in (b) second half of the input pattern is modified, whereas in (c) first half of the input pattern is modified.

## 5    Head-Rotation Dynamics

The difference in the spike rate of the two output neurons generated by the network indicates angular deviation between the head-aim and the object detected (Figure 3b). In order to orient the tracking-head in the direction of maximum sound intensity, the head-aim is rotated by a pre-specified angle for every spike, defined as the Angle of Rotation (AoR). AoR is a function of the operating spike frequency of the network and the nature of input source coding (Poisson/Uniform). It is an engineered parameter obtained by minimizing RMS error during constant angle tracking. We have provided AoR values for a range of SNN frequency operation also ensuring that AoR chosen can be achieved by commercial motors (Details in Supplementary Materials).

In a biological system, not every spike will necessarily cause a head turn as information transmission through the neuromuscular junction is stochastic in nature. To model this, we specify that an AoR turn is executed according to a probability model given as

$$\dot{\theta} = \left[ (s_l - s_r)p_i - (r_l - r_r)p_j \right] AoR \tag{4}$$

where $s_{l,r}$ is 1 if spike is issued in left (or right) IC neuron and 0 otherwise and $r_{l,r}$ is 1 if a spike is issued in left (or right) AVCN neurons. $p_i$ and $p_j$ are Bernoulli random variables (with mean values $\langle p_i \rangle = 0.5$ and $\langle p_j \rangle = 0.0005$) denoting the probability that an output and input spike causes a turn respectively. The direction and amplitude of the turn is naturally encoded in the spike rates of output and input neurons. The sign of $r_l - r_r$ is opposite to $s_l - s_r$ as a higher spike rate in right (or left) AVCN layer implies higher spike rate in left (or right) IC layer and hence they should have same causal effect on head aim. We have assigned our 'artificial bat' a fixed speed of $15\,\text{mph}$ ($6.8\,\text{m/s}$) consistent with biologically observed bat speeds [12].

## 6    Noise Modeling

In order to study the impact of realistic noisy environments on the performance of our network, we incorporate noise in our simulations by adding a slowly varying component to the source sound Intensity, $I_s$. Hence, (3) is modified as

$$\lambda_{r/l} = k(I_{d,r/l} + n) \tag{5}$$

where $n$ is obtained by low-pass filtering uniform noise. Also note that for Poisson input source encoding, since we are sampling a random signal for a fixed duration, large variations in the stimulus spike count is possible for the same values of input intensity. We will study the effect of the above additive uniform noise for both encoding schemes.

## 7    Simulation Results

We first show our system's response to a stair-case input, i.e., the source is moving along a circle with the 'bat' fixed at the center, but free to turn along the central axis (Figure 5). It can be seen that the network performs reasonably well in tracking the moving source within a second.

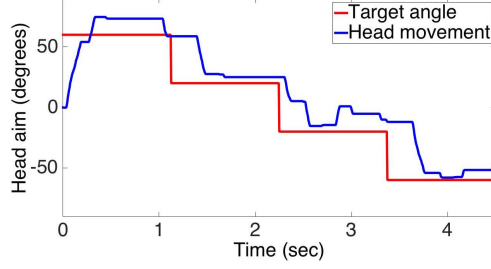

Figure 5: Response of the azimuth tracking network for time varying staircase input for Poisson input encoding at $10\,\mathrm{Hz}$ operating frequency.

We now study the step response of our SNN based azimuth tracking system for both uniform and Poisson source encoding schemes at various operating frequencies of the network, with and without additive noise (Figure 6). To quantify the performance of our system, we report the following two metrics: (a) Time of Arrival (ToA) which is the first time when the head aim comes within $5\%$ of target head aim (source angle); and (b) RMS error in head aim measured in the interval $[\mathrm{ToA}, 4.5\,\mathrm{s}]$.

At $t = 0$, the network starts tracking a stationary source placed at $-60°$; the ToA is $\sim 1\,\mathrm{s}$ in all cases, even in the presence of $50\%$ additive noise. The trajectories for $1\,\mathrm{kHz}$ Poisson encoding is superior to that corresponding to its low frequency counterpart. At low frequencies, there are not enough spikes to distinguish between small changes in angles as the receiver's sampling period is only $300\,\mathrm{ms}$. It is possible to tune the system to have much better RMS error by increasing the sampling period or decreasing AoR, but at the cost of larger ToA. Our design parameters are chosen to mimic the biologically observed ToA while minimizing the RMS error [13]. We observed that uniform source encoding performs better than Poisson encoding in terms of average jitter after ToA, as there is no sampling noise present in former.

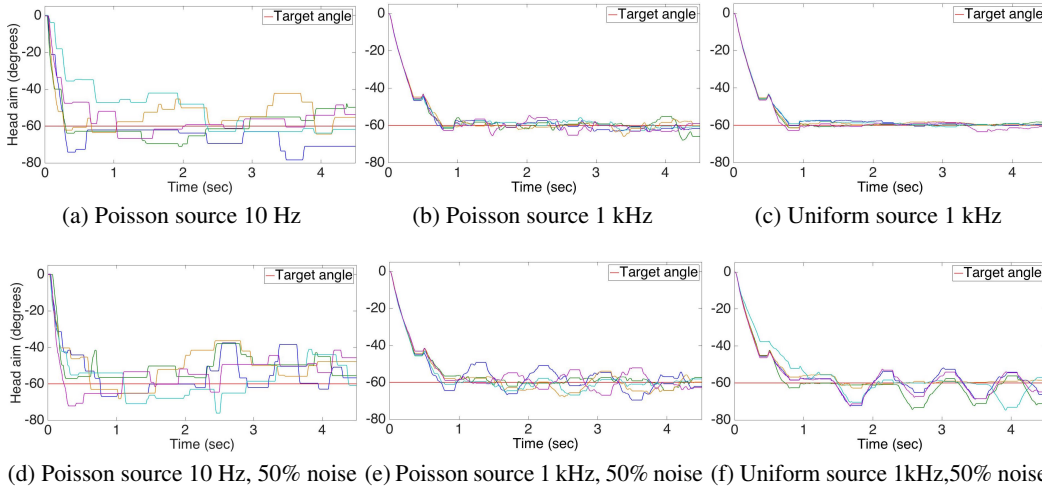

(a) Poisson source 10 Hz      (b) Poisson source 1 kHz      (c) Uniform source 1 kHz

(d) Poisson source 10 Hz, 50% noise   (e) Poisson source 1 kHz, 50% noise   (f) Uniform source 1kHz,50% noise

Figure 6: Step response of our SNN based azimuth tracking system, for five different exemplary tracks for different input signal encoding schemes, network frequencies and input noise levels. At $t = 0$, the network starts tracking a stationary source placed at $-60°$. The time taken to reach within $5\%$ of the target angle, denoted as Time of Arrival (ToA), is $\sim 1\,\mathrm{s}$ for all cases.

We expect RMS error to increase with decrease in operation frequency and increase in percentage channel noise. Figure 7a clearly shows this behavior for uniform source encoding. With no additive noise (pink label), the RMS error decreases with increase in frequency. Although RMS error remains almost constant with varying noise level for $10\,\mathrm{Hz}$ (in terms of median error and variance in error), it clearly increases for $1\,\mathrm{kHz}$ case. This can be attributed to the fact that since our 'artificial bat' moves whenever a spike occurs, at lower frequency, the network itself filters the noise by using it's slowly varying nature and averaging it. At higher frequencies, this averaging effect is reduced making

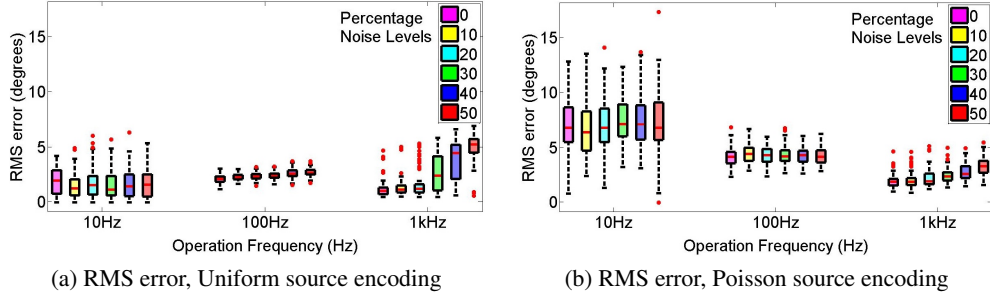

(a) RMS error, Uniform source encoding       (b) RMS error, Poisson source encoding

Figure 7: a) RMS error in head aim for Uniform source encoding measured after the ToA during tracking a constant target angle in response to varying noise levels. At zero noise, increasing the frequency improves performance due to fine-grained decisions. However, in the presence of additive noise, increasing the frequency worsens the RMS error, as more error-prone decisions are likely. b) RMS error with Poisson source encoding: at zero noise, an increase in operation frequency reduces the RMS error but compared to Figure 7a, the performance even at 1 kHz is unaffected by noise.

the trajectory more susceptible to noise. A trade-off can be seen for $50\%$ noise (red label), where addition of noise is more dominating and hence the system performs worse when operated at higher frequencies. Figure 7b reports the frequency dependence of the RMS error for the Poisson encoding scheme. Performance improves with increase in operation frequency as before, but the effect of added noise is negligible even at $50\%$ additive noise, showing that this scheme is more noise resilient. It should however be noted that performance of Poisson is at best equal to that of uniform encoding.

# 8 Performance Evaluation

To test the navigational efficiency of our design, we test its ability to track down targets while avoiding obstacles on its path in a 2D arena ($120 \times 120$ m). The target and obstacles are modeled as a point sources which emit fixed intensity sound signals. Net detected intensity due to these sources is calculated as a linear superposition of all the intensities by modifying (2) as

$$I_d = \sum_t \frac{I_t}{R_t^2} \times 10^{-1+cos(\alpha_t)} + \sum_o \frac{I_o}{R_o^2} \times 10^{-1+cos(\pi+\alpha_o)} \qquad (6)$$

where subscript $t$ refers to targets and $o$ to obstacles. Choosing the effective angle of the obstacles as $\pi + \alpha_o$ has the effect of steering the 'bat' 180° away from the obstacles. There are approximately 10 obstacles for every target in the arena placed at random locations.

Neurobiological studies have identified a range detection network which determines the modulation of bat's voice signal depending on the distance to its prey [1]. Our model does not include it; we replace the process of the bat generating sound signals and receiving echoes after reflection from surrounding objects, by the targets and obstacles themselves emitting sound signals isotropically. It is known that the bat can differentiate between prey and obstacles by detecting slight differences in their echoes [14]. This ability is aided by specialized neural networks in bat's nervous system. Since our 'artificial bat' employs a network which detects azimuth information, we model it artificially.

To benchmark the efficiency of our SNN based navigation model, we compare it with the performance of a particle that obeys standard second-order PID control system dynamics governed by the equation

$$\frac{d^2(\theta - \theta_t)}{dt^2} + k_1 \frac{d(\theta - \theta_t)}{dt} + k_2(\theta - \theta_t) = 0 \qquad (7)$$

The particle calculates a target angle $\theta_t$, which is chosen to be the angle at which the net detected intensity calculated using (6) is a maximum. This calculation is performed periodically (every $450$ ms, SNN sampling period). The above PID controller thus tries to steer the instantaneous angle of the particle $\theta$ towards the desired target angle. The parameters $k_1$ and $k_2$ (Refer Supplementary material) have been chosen to match the rise-time and overshoot characteristics of the SNN-model.

In order to compare performance under noisy conditions we add $50\%$ slow varying noise to the sound signal emitted by targets and obstacles as explained in Section 6. We simulate the trajectory for

18 s (40 sampling periods of the bat) and report the number of successful cases where the particle 'reached' the target without 'running' into any obstacles (i.e., particle-target separation was less than 2 m and particle-obstacle separation was always more than 2 m). Table 1 summarizes the results for these scenarios - the SNN model operating at 1000 Hz has significantly higher % Success and comparable average success time, though the PID particle is highly efficient in avoiding obstacles.

Table 1: Performance Validation Results

|  | SNN 1 kHz | SNN 100 Hz | SNN 10 Hz | PID |
|---|---|---|---|---|
| % Success | 68 | 66.2 | 28.4 | 29.13 |
| % No-collision | 2.4 | 3.6 | 21.6 | 60.86 |
| % Obstacle | 29.6 | 30.2 | 50 | 10 |
| Avg. success time (sec) | 6.27 | 6.66 | 6.68 | 5.08 |

To compare the computational effort of these approaches, we define 'number of decisions' as number of changes made in head aim while navigating. The SNN model utilizes $220X$ times less number of decisions while suffering $< 1\%$ decrease in % Success and a $31.5\%$ increase in average success time as compared to PID particle. Our network when operated at 100Hz (1000Hz) still retains its efficiency in terms of decision making as it incurs 20 (10) times lesser decisions respectively, as compared to the PID particle while achieving much higher % Success. A closer look at the trajectories traced by the bat and the PID particle shows that the PID particle has a tendency to get stuck in local maxima of sound intensity space, explaining why it shows high % No-collision but poor foraging (Figure 8b).

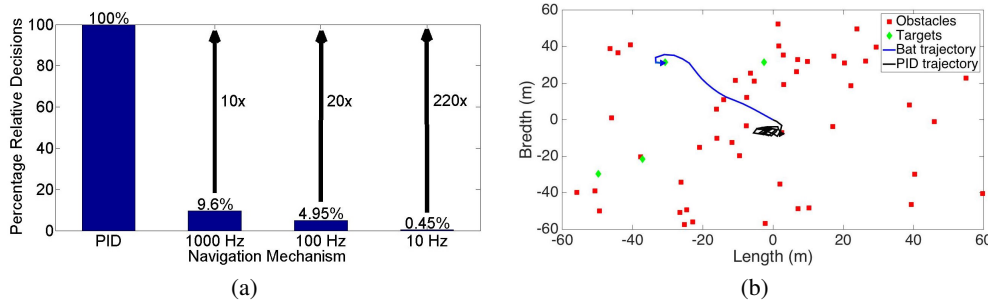

(a)   (b)

Figure 8: a) At $50\%$ slowly-varying additive noise, our network requires up to 220x lesser spike-triggered decisions, while suffering less than $1\%$ loss in performance compared to a PID control algorithm. Superior performance can be obtained at a higher spike rate of $\sim 100\,\text{Hz}$ and $\sim 1000\,\text{Hz}$, but even the accelerated networks requires 20x and 10x lesser decisions respectively (a decision corresponds to a change in the head aim). b) Exemplary tracks traced by the SNN (blue) and the PID particle (black) in a forest of obstacles (red dots) with sparse targets (green dots).

## 9   Conclusion

We have devised an azimuth detection spiking neural network for navigation and target tracking, inspired by the echolocating bat. Our network can track large angular targets ($60°$) within $1\,\text{sec}$ with a $10\%$ mean RMS error, capturing the main features of observed biological behavior. Our network performance is highly resilient to additive noise in the input and exhibits efficient decision making while navigating and tracking targets in a forest of obstacles. Our SNN based model that mimics several aspects of information processing of biology requires less than $200X$ decisions while suffering $< 1\%$ loss in performance, compared to a standard proportional-integral-derivative based control. We thus demonstrate that appropriately engineered neural information processing systems can outperform conventional control algorithms in real-life noisy environments.

**Acknowledgments**

This research was supported in part by the CAMPUSENSE project grant from CISCO Systems Inc.

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
