[Supplementary Material · nips_supp_final.pdf]

# Efficient and Robust Spiking Neural Circuit for Navigation Inspired by Echolocating Bats

**Pulkit Tandon, Yash H. Malviya**
Indian Institute of Technology, Bombay
pulkit1495,yashmalviya94@gmail.com

**Bipin Rajendran**
New Jersey Institute of Technology
bipin@njit.edu

## Supplementary Material

### Angle of Rotation

Figure 1(a) shows the dependence of Angle of Rotation (AoR) on the operating frequency for Uniform and Poisson sources. This figure indicate one particular value which gives low RMS error for a particular frequency when subjected to step function response. Hence, using this interpolation relation, our SNN can be engineered to work at a desired average spike frequency of the network. Figure 1(b) shows a histogram of theta length (calculated analogous to the run length in a coin toss). It shows that theta-length has a mode of $0.39°$ and thus AoR could be realized using commercial motors without significant error even though the AoR is small.

(a)　　　　　　　　　　　　　　　(b)

Figure 1: a) In our model, the Angle of Rotation (AoR) per spike is chosen to be inversely proportional to the operating frequency. b) The distribution of theta length in our simulations (similar to the run length in a coin toss) has a mode value $0.39°$ for an exemplary $1\,\text{kHz}$ Poisson network.

### Matching PID system to SNN characteristics

The values of $k_1$ and $k_2$ in equation (7) in text were determined by matching peak time ($T_p$) and percent-overshoot ($\eta$) with the observed values for artificial bat. Thus, using standard notation for second-order systems,

$$T_p = \frac{\pi}{\omega_n \sqrt{1 - \zeta^2}} = 0.75 \qquad \eta = 100 \exp\left(\frac{-\zeta}{\sqrt{1 - \zeta^2}}\pi\right) = 10$$

and hence values of $k_1$ and $k_2$ are given by

$$k_1 = 2\zeta\omega_n = 6.12 \qquad k_2 = \omega_n^2 = 26.9.$$

**Simulation parameters**

The following tables specifies parameter values used in our simulation model:

Table 1: AEIF Neuronal Parameters used in Simulation for neuron types RS and CH

| Parameter | RS | CH | Parameter | RS | CH |
|---|---|---|---|---|---|
| C | 200pF | 200pF | $\tau_w$ | 30ms | 120ms |
| $g_L$ | 10ns | 10ns | a | 2ns | 2ns |
| $E_L$ | -70mV | -58mV | b | 0pA | 100pA |
| $\Delta_T$ | 2mV | 2mV | $V_r$ | -58mV | -46mV |
| $V_T$ | -50mV | -50mV | | | |

Table 2: Values of synaptic strengths for sub network $n$, where $n$ is an integer from 1 to 16. $W_{i,j}$ denotes the synapse from neuron $i$ to neuron $j$.

| Synapse | Value | Synapse | Value | Synapse | Value |
|---|---|---|---|---|---|
| $W_{1,3}$ | $80 + 2n$ | $W_{1,4}$ | $-80$ | $W_{2,3}$ | $-80$ |
| $W_{2,4}$ | $80 + 2n$ | $W_{3,5}$ | $-20$ | $W_{3,6}$ | $55$ |
| $W_{3,7}$ | $-8$ | $W_{3,8}$ | $65$ | $W_{4,5}$ | $65$ |
| $W_{4,6}$ | $-8$ | $W_{4,7}$ | $65$ | $W_{4,8}$ | $-8$ |
| $W_{5,7}$ | $-7$ | $W_{5,8}$ | $-7$ | $W_{6,7}$ | $-7$ |
| $W_{6,8}$ | $-7$ | $W_{7,8}$ | $-10$ | $W_{8,7}$ | $-10$ |

Table 3: Other simulation parameters

| $k$ (10 Hz) | 1 | $k_1$ | 6.12 |
|---|---|---|---|
| $k$ (100 Hz) | 10 | $k_2$ | 26.9 |
| $k$ (1 kHz) | 100 | $v$ | 6.8 m/s |