[Reviews · NeurIPS 2016]

Reviewer 1

Summary

The paper descries a navigation system based on a spiking neural network (SNN) for azimuth detection, inspired by bat echolocation. The system mimics the biological system used by bats, it takes a left and right sound signals and returns a head-aim (angle in 2d) directed towards maximum intensity. The system was successfully tested on the task of tracking an object rotating around the input sensors, and on the task of tracking a moving prey in a 2D plan with obstacles. The system is also shown to outperform a PID controller in the latter task.

Qualitative Assessment

The paper is clear and well-written. The described system seems to work well on simulated tasks. I have the following questions: 1) How much fine-tunning was applied to the parameters of the PID controller? It is surprising to that it didn't work well on a simple task of tracking the source of a sound in 2D. It would be nice to provide explanations for that. Would you say that PID would work equally well as your system given good fine-tunning? Or do you suggest to use your system for real applications, such as robotics? 2) It would be nice to see a comparaison to a simple neural net trained with real data. I suppose it would perform better than the biologically inspired net. If the goal is to provide a tool that will be used in practice, then it's important to do this type of comparaisons. 3) Have you tested the system on 3D tracking? 4) What did we learn from this research? That the proposed system is a good model (approximation) of the biological system?

Confidence in this Review

2-Confident (read it all; understood it all reasonably well)


Reviewer 2

Summary

This paper proposes a spiking neural network (SNN) design to detect and track the azimuth angle of a target, which emits sound waves, while avoiding obstacles. The design is inspired by the echolocation system of a specific type of bat. The system’s performance (tracking a target angle) is compared to a conventional control system with comparable results while requiring less computational effort.

Qualitative Assessment

This paper is very well written, interesting to read and, to me, it makes a relevant contribution to the area of bio-inspired detection and control systems, in particular, via SNN designs. I only have a few minor comments and questions: 1) The sampling times and duration (every 450 ms, and 300 ms) given after equation (3), are these inspired by the bat, or chosen based on some technical considerations? Please provide some insight. 2) AVCN does not seem to be introduced. 3) In Figure 3(b), top graph, it is not clear to me, what outputs (O_i^j) these signals correspond to. Can you make this precise? 4) In Figure 3(a), what is the role of the outputs O_1^j and O_4^j? Is this explained in the text? (I might have missed this...). If I understand correctly, the outputs of the network are used in (4); the authors might want to make this connection clearer. 5) How is the SNN trained? 6) At the end of Section 4, it is not clear what the authors mean by “the network to exhibit proportional-integral-derivative controller like behavior”. If they refer to the property that the network has different outputs depending on the input history, the more appropriate terminology might be to say that the network can represent internal (control) states or controllers with memory. Proportional-integral-derivative (PID) control is a specific type of controller that produces a controller output that is a weighted summation of the control error, the integral of that error, and its derivative. I don’t see this specific behavior in Figure 4. (If I missed this, please explain.) 7) In the simulation results, what causes the roughly 1 sec that it takes to detect the target angle (e.g. in Figs. 5 and 6)? Is this determined by the processing speed of the network? (In this case, I would expect it to decrease when the frequency increases to 1kHz, which does not seem to be the case.) Anyway, it seems that 1 sec is rather long to associate it with the processing time... I’d appreciate if the authors could explain. 8) I don’t understand why the authors call the system (7) a PID control system. As pointed out above, PID is a specific control strategy. In (7), I don’t recognize an integral term, for instance (unless one considers this as tracking an angular velocity, which does not seem to be the case, however, as the angle seems to be the main target variable here). (7) looks like a state-feedback system to me; or one could possibly call this PD (without the I). Please explain what you had in mind here. 9) What does sampling period of the SNN (e.g. 450 ms) mean? Does this mean that the SNN computes a new output every 450 ms? 10) In table 1, what is the difference between “no collision” and “obstacle”?

Confidence in this Review

1-Less confident (might not have understood significant parts)


Reviewer 3

Summary

Authors demonstrate a spiking neural circuit for azimuth angle detection inspired by the echolocation circuits of the Horseshoe bat Rhinolophus ferrumequinum. They next use it to devise a model for navigation and target tracking, capturing several key aspects of information transmission in biology. Through their proposal, authors can track angular targets (60◦) within 1 sec with a 10% RMS error.

Qualitative Assessment

Things to do to improve paper quality: 1) In section 4 authors mention that their SNN consists of 16 sub-networks whose spike outputs are summed up to generate the network output.Why 16 sub-networks? Please explain. 2) In section 4 authors talk about AVCN neurons. Please explain. 3) Can you compare your proposal with other (standard and non standard) methods reported in literature performing the same task. I you cannot, explain why this is not possible.

Confidence in this Review

2-Confident (read it all; understood it all reasonably well)


Reviewer 4

Summary

The authors design a biologically inspired neural circuit for controlling the angle of a bat in relation to targets and obstacles detected with echolocation. The proposed neural circuit is modelled as a spiking neural network (SNN) and estimates the angle to objects. Experimental evaluation looks at spike rate in the presence of environment noise and show that a higher spike rate is not necessarily best when the environment is very noisy. The method is also compared to a PID controller on a foraging task and performs better with less computation.

Qualitative Assessment

I found this paper difficult to judge because I'm unsure of what the intended contribution is. From my reading of it, the authors propose a SNN model that is based on available literature of how bats track target angles. Design decisions throughout the paper are chosen to make the model more bat like. The intent is that by designing this system the authors are able to show that biologically based systems can outperform a classic control algorithm while being more computationally efficient. In general, the technical quality of the paper is strong with several experiments, thorough analysis of them, and the method described clearly. The authors first look at a parameter of the model (spike rate) and show that a high spike rate is usually good it can be problematic when environment noise is high. They next compare their model to a PID controller for a foraging and target avoidance task. One concern I have about this experiment is that the PID controller is tuned to match characteristics of the proposed model. If the comparison is being made on % success, then the PID controller should be tuned to maximize success. To the best of my knowledge, this is the first work that has modelled azimuth angle detection with a Spiking NN. The proposed model outperforms a PID controller for a foraging task which shows that, in this instance, biological systems can outperform a classic control algorithm. My biggest concern with the paper is the generality of its contribution. While it is interesting that a biologically inspired network can out perform a PID controller, I'm not sure what the main contribution is aside from azimuth angle detection that mimics what is known about how bats solve this problem. If nothing more can be said about how this could relate to modelling other tasks in other animals or applying SNNs to other problems then the impact seems very low. The paper is well written. In most places it is very clear except for some details. I'm not sure why 16 clone networks are used. The paragraph describing this (100 - 105) is vague and I can only assume it is a form of averaging although I'm not sure why this is needed. I'm also unsure about how the network parameters were chosen. There is a statement that they were chosen so the network could work with different spike rates (Line 111) but wouldn't it make more sense to find the parameters that worked best for each spike rate and use those? Minor comments: In general the paper uses a lot of acronyms which are hard to keep in memory while reading. If the authors think this is unavoidable then it would be useful to periodically redefine them. This would enhance the readability of the paper. [Update: I read the author's response and am satisfied with answer on how PID was tuned. After reading other reviews and author's response I raised my novelty score.]

Confidence in this Review

1-Less confident (might not have understood significant parts)


Reviewer 5

Summary

The paper describes an interesting biologically inspired spiking neural network (SNN) model that tires to emulate the ability of bats to echolocate their prey/obstcles. The paper provides a description of the SNN model that has been developed and explains how the SNN can be incorporated into a head rotation model that works well in the presence of noise. The authors present a short evaluation of their model comapred to a relatively simple PID-controller alternative.

Qualitative Assessment

The paper was stimulating and interesting to read though (in hard-copy form) it was rather difficult to follow at times. This is mainly because some of the graphs and figures are too small. For example Figure 6 - each of the six graphs in this figure have incomplete legends/captions and even if this is corrected the current size of the graphs makes reading any particular coloured plotline almost impossible. Also, Figure 8(b) - this plot is so small that it is impossible to properly see the Bat or PID trajectories it tries to present. There are a few other, largely cosmetic, typographical problems: "cardoid" in Section 2 paragraph 3; "relies it on" opening sentence Section 3; "AVCN" acronym first use not expanded Section 4 paragraph 1; "We observer" Section 7 paragraph 3; each of which can be trivially attended to and corrected by the authors. Slightly more troubling though was the lack of detail at certain points eg.: Regarding the "parameter space exploration" that was carried out to determine synapse weightings according to Section 4 paragraph 3. It would be good to know how the space of potential synapse weights was explored. Section 8 paragrpah 2, the authors point out that their SNN addresses azimuth angle detection rather than discriminating sound signals from prey/obstacles and that they therefore "model it artificially" but do not clearly explain how.

Confidence in this Review

2-Confident (read it all; understood it all reasonably well)


Reviewer 6

Summary

The paper devises an azimuth detection spiking neural network for navigation and target tracking. The designed network can track large angular targets (60 degree) within 1 sec with a 10% mean RMS error. The simulation demonstrates that the network performance is highly resilient to additive noise in the input and exhibits efficient decision making while navigating and tracking targets in a forest of obstacles. The demonstration implies that appropriately engineered neural information processing systems can outperform conventional control algorithms.

Qualitative Assessment

The paper looks a good system design and experiment work. The only concern is that the paper does not have enough theoretical analysis of the experimental results.

Confidence in this Review

1-Less confident (might not have understood significant parts)